# Robustness Verification of Tree-based Models

**Hongge Chen**[*,1]    **Huan Zhang**[*,2]    **Si Si**[3]    **Yang Li**[3]    **Duane Boning**[1]    **Cho-Jui Hsieh**[2,3]

[1]Department of EECS, MIT
[2]Department of Computer Science, UCLA
[3]Google Research
chenhg@mit.edu, huan@huan-zhang.com, sisidaisy@google.com
liyang@google.com, boning@mtl.mit.edu, chohsieh@cs.ucla.edu

[*]Hongge Chen and Huan Zhang contributed equally.

## Abstract

We study the robustness verification problem for tree based models, including decision trees, random forests (RFs) and gradient boosted decision trees (GBDTs). Formal robustness verification of decision tree ensembles involves finding the exact minimal adversarial perturbation or a guaranteed lower bound of it. Existing approaches find the minimal adversarial perturbation by a mixed integer linear programming (MILP) problem, which takes exponential time so is impractical for large ensembles. Although this verification problem is NP-complete in general, we give a more precise complexity characterization. We show that there is a simple linear time algorithm for verifying a single tree, and for tree ensembles the verification problem can be cast as a max-clique problem on a multi-partite graph with bounded boxicity. For low dimensional problems when boxicity can be viewed as constant, this reformulation leads to a polynomial time algorithm. For general problems, by exploiting the boxicity of the graph, we develop an efficient multi-level verification algorithm that can give tight lower bounds on robustness of decision tree ensembles, while allowing iterative improvement and any-time termination. On RF/GBDT models trained on 10 datasets, our algorithm is hundreds of times faster than a previous approach that requires solving MILPs, and is able to give tight robustness verification bounds on large GBDTs with hundreds of deep trees.

## 1   Introduction

Recent studies have demonstrated that neural network models are vulnerable to adversarial perturbations—a small and human imperceptible input perturbation can easily change the predicted label [37, 17, 6, 15]. This has created serious security threats to many real applications so it becomes important to formally verify the robustness of machine learning models. Usually, the robustness verification problem can be cast as finding the minimal adversarial perturbation to an input example that can change the predicted class label. A series of robustness verification algorithms have been developed for neural network models [21, 38, 43, 42, 41, 47, 16, 35], where efficient algorithms are mostly based on convex relaxations of nonlinear activation functions of neural networks [32].

We study the robustness verification problem of tree-based models, including a single decision tree and tree ensembles such as random forests (RFs) and gradient boosted decision trees (GBDTs). These models have been widely used in practice [12, 22, 46] and recent studies have demonstrated that both RFs and GBDTs are vulnerable to adversarial perturbations [20, 13, 9]. It is thus important to develop a formal robustness verification algorithm for tree-based models. Robustness verification requires computing the minimal adversarial perturbation. [20] showed that computing minimal adversarial perturbation for tree ensemble is NP-complete in general, and they proposed a Mixed-Integer Linear Programming (MILP) based approach to compute the minimal adversarial perturbation. Although

exact verification is NP-hard, in order to have an efficient verification algorithm for real applications we seek to answer the following questions:

- Do we have polynomial time algorithms for exact verification under some special circumstances?
- For general tree ensemble models with a large number of trees, can we efficiently compute meaningful lower bounds on robustness while scaling to large tree ensembles?

In this paper, we answer the above-mentioned questions affirmatively by formulating the verification problem of tree ensembles as a graph problem. First, we show that for a single decision tree, robustness verification can be done exactly in linear time. Then we show that for an ensemble of $K$ trees, the verification problem is equivalent to finding the maximum cliques in a $K$-partite graph, and the graph is in a special form with boxicity equal to the input feature dimension. Therefore, for low-dimensional problems, verification can be done in polynomial time with maximum clique searching algorithms. Finally, for large-scale tree ensembles, we propose a multiscale verification algorithm by exploiting the boxicity of the graph, which can give tight lower bounds on robustness. Furthermore, it supports any-time termination: we can stop the algorithm at any time to obtain a reasonable lower bound given a computation time constraint. Our proposed algorithm is efficient and is scalable to large tree ensemble models. For instance, on a large multi-class GBDT with 200 trees robustly trained (using [9]) on the MNIST dataset, we obtained 78% verified robustness accuracy on test set with maximum $\ell_\infty$ perturbation $\epsilon = 0.2$ and the time used for verifying each test example is 12.6 seconds, whereas the MILP method uses around 10 min for each test example.

## 2 Background and Related Work

**Adversarial Robustness** For simplicity, we consider a multi-class classification model $f : \mathbb{R}^d \to \{1, \ldots, C\}$ where $d$ is the input dimension and $C$ is number of classes. For an input example $x$, assuming that $y_0 = f(x)$ is the correct label, the **minimal adversarial perturbation** is defined by

$$r^* = \min_\delta \|\delta\|_\infty \quad \text{s.t.} \quad f(x + \delta) \neq y_0. \tag{1}$$

Note that we focus on the $\ell_\infty$ norm measurement in this paper which is widely used in recent studies [25, 43, 5]. Exactly solving (1) is usually intractable. For example, if $f(\cdot)$ is a neural network, (1) is non-convex and [21] showed that solving (1) is NP-complete for ReLU networks.

**Adversarial attacks** are algorithms developed for finding a feasible solution $\bar{\delta}$ of (1), where $\|\bar{\delta}\|_\infty$ is an *upper bound* of $r^*$. Many algorithms have been proposed for attacking machine learning models [17, 23, 6, 25, 10, 11, 18, 3, 13, 28, 24, 45]. Most practical attacks cannot guarantee to reach the minimal adversarial perturbation $r^*$ due to the non-convexity of (1). Therefore, attacking algorithms cannot provide any formal guarantee on model robustness [1, 40].

On the other hand, **robustness verification algorithms** are designed to find the exact value or a *lower bound* of $r^*$. An exact verifier needs to solve (1) to the global optimal, so typically we resort to relaxed verifiers that give lower bounds. After a verification algorithm finds a lower bound $\underline{r}$, it guarantees that no adversarial example exists within a radius $\underline{r}$ ball around $x$. This is important for deploying machine learning algorithms to safety-critical applications such as autonomous vehicles or aircraft control systems [21, 19].

For verification, instead of solving (1) we can also solve the following **decision problem of robustness verification**

$$\textit{Does there exist an } x' \in \text{Ball}(x, \epsilon) \quad \textit{such that } f(x') \neq y_0? \tag{2}$$

In our setting $\text{Ball}(x, \epsilon) := \{x' : \|x' - x\|_\infty \leq \epsilon\}$. If we can answer this decision ("yes"/"no") problem, a binary search can give us the value of $r^*$, so the complexity of (2) is in the same order of (1). Furthermore, solving (1) using an approximation algorithm (with answer "unknown" allowed) can lead to a lower bound of $r^*$, which is useful for verification. The decision version is also widely used in the verification community since "verified accuracy under $\epsilon$ perturbation" is an important metric, which is defined as the portion of test samples that the answers to (2) are "no". Verification methods for neural networks have been studied extensively in the past few years [43, 44, 42, 47, 35, 16, 36].

**Adversarial Robustness of Tree-based Models** Unlike neural networks, decision-tree based models are non-continuous step functions, and thus existing neural network verification techniques cannot be directly applied. In [2], a single decision tree was verified to evaluate the robustness of reinforcement learning policies. For tree ensembles, [20] showed that solving (1) for general tree

ensemble models is NP-complete, so no polynomial time algorithm can compute $r^*$ for arbitrary trees unless P=NP. A Mixed Integer Linear Programming (MILP) algorithm was thus proposed in [20] to compute (1) in exponential time. Recently, [14] and [33] verify the robustness of tree ensembles using an SMT solver, which is also NP-complete in its natural formulation. Additionally, an approximate bound for tree ensembles was proposed recently in [39] by directly combining the bounds of each tree together, which can be seen as a special case of our proposed method.

On the other hand, robustness can be empirically evaluated through adversarial attacks [27]. Some hard-label attacking algorithms for neural networks, including the boundary attack [3] and OPT-attack [13], can be applied to tree based models since they only require function evaluation of the non-smooth (hard-label) decision function $f(\cdot)$. These attacks computes an upper bound of $r^*$. In contrast, our work focuses on efficiently computing a tight lower bound of $r^*$ for ensemble trees.

# 3 Proposed Algorithm

The exact verification problem of tree ensemble is NP-complete by its nature, and here we propose a series of efficient verification algorithms for real applications. First, we will introduce a linear time algorithm for exactly computing the minimal adversarial distortion $r^*$ for verifying a single decision tree. For an ensemble of trees, we cast the verification problem into a max-clique searching problem in K-partite graphs. For large-scale tree ensembles, we then propose an efficient multi-level algorithm for verifying an ensemble of decision trees.

## 3.1 Exactly Verifying a Single Tree in Linear Time

Although computing $r^*$ for a tree ensemble is NP-complete [20], we show that a **linear time** algorithm exists for finding the minimum adversarial perturbation and computing $r^*$ for a single decision tree. We assume the decision tree has $n$ nodes and the root node is indexed as $0$. For a given example $x = [x_1, \ldots, x_d]$ with $d$ features, starting from the root, $x$ traverses the decision tree model until reaching a leaf node. Each internal node, say node $i$, has two children and a univariate feature-threshold pair $(t_i, \eta_i)$ to determine the traversal direction—$x$ will be passed to the left child if $x_{t_i} \leq \eta_i$ and to the right child otherwise. Each leaf node has a value $v_i$ corresponding to the predicted class label for a classification tree, or a real value for a regression tree.

Conceptually, the main idea of our single tree verification algorithm is to compute a $d$-dimensional box for each leaf node such that any example in this box will fall into this leaf. Mathematically, the node $i$'s box is defined as the Cartesian product $B^i = (l_1^i, r_1^i] \times \cdots \times (l_d^i, r_d^i]$ of $d$ intervals on the real line. By definition, the root node has box $[-\infty, \infty] \times \cdots \times [-\infty, \infty]$ and given the box of an internal node $i$, its children's boxes can be obtained by changing only one interval of the box based on the split condition $(t_i, \eta_i)$. More specifically, if $p, q$ are node $i$'s left and right child node respectively, then we set their boxes $B^p = (l_1^p, r_1^p] \times \cdots \times (l_d^p, r_d^p]$ and $B^q = (l_1^q, r_1^q] \times \cdots \times (l_d^q, r_d^q]$ by setting

$$(l_t^p, r_t^p] = \begin{cases} (l_t^i, r_t^i] & \text{if } t \neq t_i \\ (l_t^i, \min\{r_t^i, \eta_i\}] & \text{if } t = t_i \end{cases}, \quad (l_t^q, r_t^q] = \begin{cases} (l_t^i, r_t^i] & \text{if } t \neq t_i \\ (\max\{l_t^i, \eta_i\}, r_t^i] & \text{if } t = t_i. \end{cases} \quad (3)$$

After computing the boxes for internal nodes, we can also obtain the boxes for leaf nodes using (3). Therefore computing the boxes for all the leaf nodes of a decision tree can be done by a depth-first search traversal of the tree with time complexity $O(nd)$.

With the boxes computed for each leaf node, the minimum perturbation required to change $x$ to go to a leaf node $i$ can be written as a vector $\boldsymbol{\epsilon}(x, B^i) \in \mathbb{R}^d$ defined as

$$\boldsymbol{\epsilon}(x, B^i)_t := \begin{cases} 0 & \text{if } x_t \in (l_t^i, r_t^i] \\ x_t - r_t^i & \text{if } x_t > r_t^i \\ l_t^i - x_t & \text{if } x_t \leq l_t^i. \end{cases} \quad (4)$$

Then the minimal distortion can be computed as $r^* = \min_{i:v_i \neq y_0} \|\boldsymbol{\epsilon}(x, B^i)\|_\infty$, where $y_0$ is the original label of $x$, and $v_i$ is the label for leaf node $i$. To find $r^*$, we check $B^i$ for all leaves and choose the smallest perturbation. This is a linear-time algorithm for exactly verifying the robustness of a single decision tree. In fact, this $O(nd)$ time algorithm is used to illustrate the concept of "boxes" that will be used later on for the tree ensemble case. If our final goal is to verify a single tree, we can have a more efficient algorithm by combining the distance computation (4) in the tree traversal procedure, and the resulting algorithm will take only $O(n)$ time. This algorithm is presented as Algorithm 3 in the Appendix.

## 3.2 Verifying Tree Ensembles by Max-clique Enumeration

Now we discuss the robustness verification for tree ensembles. Assuming the tree ensemble has $K$ decision trees, we use $S^{(k)}$ to denote the set of leaf nodes of tree $k$ and $m^{(k)}(x)$ to denote the function that maps the input example $x$ to the leaf node of tree $k$ according to its traversal rule. Given an input example $x$, the tree ensemble will pass $x$ to each of these $K$ trees independently and $x$ reaches $K$ leaf nodes $i^{(k)} = m^{(k)}(x)$ for all $k = 1, \ldots, K$. Each leaf node will assign a prediction value $v_{i^{(k)}}$. For simplicity we start with the binary classification case, with $x$'s original label being $y_0 = -1$ and we want to turn it into $+1$. For binary classification the prediction of the tree ensemble is computed by $\mathrm{sign}(\sum_k v_{i^{(k)}})$, which covers both GBDTs and random forests, two widely used tree ensemble models. Assume $x$ has a label $y_0 = -1$, which means $\mathrm{sign}(\sum_k v_{i^{(k)}}) < 0$ for $x$, and our task is to verify if the sign of the summation can be flipped within $\mathrm{Ball}(x, \epsilon)$.

We consider the decision problem of robustness verification (2). A naive analysis will need to check all the points in $\mathrm{Ball}(x, \epsilon)$ which is uncountably infinite. To reduce the search space to finite, we start by defining some notation: let $\mathbb{C} = \{(i^{(1)}, \ldots, i^{(K)}) \mid i^{(k)} \in S^{(k)}, \ \forall k = 1, \ldots, L\}$ to be all the possible tuples of leaf nodes and let $\mathcal{C}(x) = [m^{(1)}(x), \ldots, m^{(K)}(x)]$ be the function that maps $x$ to the corresponding leaf nodes. Therefore, a tuple $C \in \mathbb{C}$ directly determines the model prediction $\sum v_C := \sum_k v_{i^{(k)}}$. Now we define a valid tuple for robustness verification:

**Definition 1.** *A tuple $C = (i^{(1)}, \ldots, i^{(K)})$ is valid if and only if there exists an $x' \in Ball(x, \epsilon)$ such that $C = \mathcal{C}(x')$.*

The decision problem of robustness verification (2) can then be written as:

$$\text{Does there exist a valid tuple } C \text{ such that } \sum v_C > 0?$$

Next, we show how to model the set of valid tuples. We have two observations. First, if a tuple contains any node $i$ with $\inf_{x' \in B^i} \{\|x - x'\|_\infty\} > \epsilon$, then it will be invalid. Second, there exists an $x$ such that $C = \mathcal{C}(x)$ if and only if $B^{i^{(1)}} \cap \cdots \cap B^{i^{(K)}} \neq \emptyset$, or equivalently:

$$(l_t^{i^{(1)}}, r_t^{i^{(1)}}] \cap \cdots \cap (l_t^{i^{(K)}}, r_t^{i^{(K)}}] \neq \emptyset, \ \ \forall t = 1, \ldots, d.$$

We show that the set of valid tuples can be represented as cliques in a graph $G = (V, E)$, where $V := \{i | B^i \cap \mathrm{Ball}(x, \epsilon) \neq \emptyset\}$ and $E := \{(i, j) | B^i \cap B^j \neq \emptyset\}$. In this graph, nodes are the leaves of all trees and we remove every leaf that has empty intersection with $\mathrm{Ball}(x, \epsilon)$. There is an edge $(i, j)$ between node $i$ and $j$ if and only if their boxes intersect. The graph will then be a $K$-partite graph since there cannot be any edge between nodes from the same tree, and thus maximum cliques in this graph will have $K$ nodes. We define each part of the $K$-partite graph as $V_k$. Here a "part" means a disjoint and independent set in the $K$-partite graph. The following lemma shows that intersections of boxes have very nice properties:

**Lemma 1.** *For boxes $B^1, \ldots, B^K$, if $B^i \cap B^j \neq \emptyset$ for all $i, j \in [K]$, let $\bar{B} = B^1 \cap B^2 \cap \cdots \cap B^K$ be their intersection. Then $\bar{B}$ will also be a box and $\bar{B} \neq \emptyset$.*

The proof can be found in the Appendix. Based on the above lemma, each $K$-clique (fully connected subgraph with $K$ nodes) in $G$ can be viewed as a set of leaf nodes that has nonempty intersection with each other and also has nonempty intersection with $\mathrm{Ball}(x, \epsilon)$, so the intersection of those $K$ boxes and $\mathrm{Ball}(x, \epsilon)$ will be a nonempty box, which implies each $K$-clique corresponds to a valid tuple of leaf nodes:

**Lemma 2.** *A tuple $C = (i^{(1)}, \ldots, i^{(K)})$ is valid if and only if nodes $i^{(1)}, \ldots, i^{(K)}$ form a $K$-clique (maximum clique) in graph $G$ constructed above.*

Therefore the robustness verification problem can be formulated as

$$\text{Is there a maximum clique } C \text{ in } G \text{ such that } \sum v_C > 0? \tag{5}$$

This reformulation indicates that the tree ensemble verification problem can be solved by an efficient maximum clique enumeration algorithm. Some standard maximum clique searching algorithms can be applied here to perform verification:

- **Finding $K$-cliques in $K$-partite graphs:** Any algorithm for finding all the maximum cliques in $G$ can be used. The classic B-K backtracking algorithm [4] takes $O(3^{\frac{m}{3}})$ time to find all

the maximum cliques where $m$ is the number of nodes in $G$. Furthermore, since our graph is a $K$-partite graph, we can apply some specialized algorithms designed for finding all the $K$-cliques in $K$-partite graphs [26, 29, 34].

- **Polynomial time algorithms exist for low-dimensional problems:** Another important property for graph $G$ is that each node in $G$ is a $d$-dimensional box and each edge indicates the intersection of two boxes. This implies our graph $G$ is with "boxicity $d$" (see [7] for detail). [7] proved that the number of maximum cliques will only be $O((2m)^d)$ and it is able to find the maximum weight clique in $O((2m)^d)$ time. Therefore, for problems with a very small $d$, the time complexity for verification is actually polynomial.

Therefore we can exactly solve the tree ensemble verification problem using algorithms for maximum cliques searching in $K$-partite graph, and its time complexity is found to be as follows:

**Theorem 1.** *Exactly verifying the robustness of a $K$-tree ensemble with at most $n$ leaves per tree and $d$ dimensional features takes* $\min\{O(n^K), O((2Kn)^d)\}$ *time.*

This is a direct consequence of the fact that the number of $K$-cliques in a $K$-partite graph with $n$ vertices per part is bounded by $O(n^K)$, and number of maximum cliques in a graph with a total of $m$ nodes with boxicity $d$ is $O((2m)^d)$. For a general graph, since $K$ and $d$ can be in $O(n)$ and $O(m)$ [31], it can still be exponential. But the theorem gives a more precise characterization for the complexity of the verification problem for tree ensembles. Based on the nice properties of maximum cliques searching problem, we propose a simple and elegant algorithm that enumerates all $K$-cliques on a $K$-partite graph with a known boxicity $d$ in Algorithm 1, and we can use this algorithm for tree ensemble verification when the number of trees or the dimension of features is small.

For a $K$-partite graph $G$, we define the set $\tilde{V} := \{V_1, V_2, \cdots, V_K\}$ which is a set of independent sets ("parts") in $G$. The algorithm first looks at any first two parts $V_1$ and $V_2$ of the graph and enumerates all 2-cliques in $O(|V_1||V_2|)$ time. Then, each 2-clique found is converted into a "pseudo node" (this is possible due to Lemma 1), and all 2-cliques form a new part $V_2'$ of the graph. Then we replace $V_1$ and $V_2$ with $V_2'$, and continue to enumerate all 2-cliques between $V_2'$ and $V_3$ to form $V_3'$. A 2-clique between $V_2'$ and $V_3$ represents a 3-clique in $V_1$, $V_2$ and $V_3$ due to boxicity. Note that enumerating all 3-cliques in a general 3-partite graph takes $O(|V_1||V_2||V_3|)$ time; thanks to boxicity, our algorithm takes $O(|V_2'||V_3|)$ time which equals to $O(|V_1||V_2||V_3|)$ only when $V_1$ and $V_2$ form a complete bipartite graph, which is unlikely in common cases. This process continues recursively until we process all $K$ parts and have only $V_K'$ left, where each vertex in $V_K'$ represents a $K$-clique in the original graph. After obtaining all $K$-cliques, we can verify their prediction values to compute a verification bound.

---

**Algorithm 1:** Enumerating all $K$-cliques on a $K$-partite graph with a known boxicity $d$

**input:** $V_1$, $V_2$, ,..., $V_K$ are the $K$ independent sets ("parts") of a $K$-partite graph

1 **for** $k \leftarrow$ 1, 2, 3, ..., $K$ **do**
2     $U_k \leftarrow \{(A_i, B^{i^{(k)}}) | i^{(k)} \in V_k, A_i = \{i^{(k)}\}\}$;
    /* $U$ is a set of tuples $(A, B)$, which stores a set of cliques and their corresponding boxes. $A$ is the set of nodes in one clique and $B$ is the corresponding box of this clique. Initially, each node in $V_k$ forms a 1-clique itself. */
3 **end**
4 CliqueEnumerate($U_1$, $U_2$, ,..., $U_K$);

5 **Function** CliqueEnumerate($U_1$, $U_2$, ,..., $U_K$)
6     $\hat{U}_{\text{old}} \leftarrow U_1$;
7     **for** $k \leftarrow$ 2, 3, ..., $K$ **do**
8        $\hat{U}_{\text{new}} \leftarrow \emptyset$;
9        **for** $(\hat{A}, \hat{B}) \in \hat{U}_{old}$ **do**
10           **for** $(A, B) \in U_k$ **do**
11              **if** $B \cap \hat{B} \neq \emptyset$ **then**
                /* A $k$-clique is found; add it as a pseudo node with the intersection of two boxes. */
12                 $\hat{U}_{\text{new}} \leftarrow \hat{U}_{\text{new}} \cup \{(A \cup \hat{A}, B \cap \hat{B})\}$;
13           **end**
14        **end**
15        $\hat{U}_{\text{old}} \leftarrow \hat{U}_{\text{new}}$;
16     **end**
17     return $\hat{U}_{\text{new}}$;
18 **end**

---

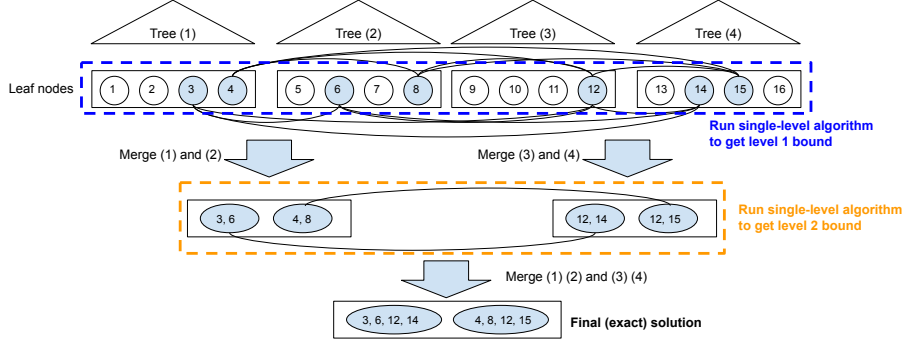

Figure 1: The proposed multi-level verification algorithm. Lines between leaf node i on tree $t_1$ and leaf node $j$ on $t_2$ indicate that their $\ell_\infty$ feature boxes intersect (i.e., there exists an input such that tree 1 predicts $v_i$ and tree 2 predicts $v_j$).

### 3.3 An Efficient Multi-level Algorithm for Verifying the Robustness of a Tree Ensemble

Practical tree ensembles usually have tens or hundreds of trees with large feature dimensions, so Algorithm 1 will take exponential time and will be too slow. We thus develop an efficient multi-level algorithm for computing verification bounds by further exploiting the boxicity of the graph.

Figure 1 illustrates the graph and how our multilevel algorithm runs. There are four trees and each tree has four leaf nodes. A node is colored if it has nonempty intersection with Ball$(x, \epsilon)$; uncolored nodes are discarded. To answer question (5), we need to compute the maximum $\sum v_C$ among all $K$-cliques, denoted by $v^*$. As mentioned before, for robustness verification we only need to compute an upper bound of $v^*$ in order to get a lower bound of minimal adversarial perturbation. In the following, we will first discuss algorithms for computing an upper bound at the top level, and then show how our multi-scale algorithm iteratively refines this bound until reaching the exact solution $v^*$.

**Bounds for a single level.** To compute an upper bound of $v^*$, a naive approach is to assume that the graph is fully connected between independent sets (fully connected $K$-partite graph) and in this case the maximum sum of node values is the sum of the maximum value of each independent set:

$$\sum_{k=1}^{|\tilde{V}|} \max_{i \in V_k} v_i \geq v^*. \tag{6}$$

Here we abuse the notation $v_i$ by assuming that each node $i$ in $V_k$ has been assigned a "pseudo prediction value", which will be used in the multi-level setting. In the simplest case, each independent set represents a single tree, $V_k = S^{(k)}$ and $v_i$ is the prediction of a leaf. One can easily show this is an upper bound of $v^*$ since any $K$-clique in the graph is still considered when we add more edges to the graph, and eventually it becomes a fully connected $K$-partite graph.

Another slightly better approach is to exploit the edge information but only between tree $t$ and $t + 1$. If we search over all the length-$K$ paths $[i^{(1)}, \ldots, i^{(K)}]$ from the first to the last part and define the value of a path to be $\sum_k v_{i^{(k)}}$, then the maximum valued path will be a upper bound of $v^*$. This can be computed in linear time using dynamic programming. We scan nodes from tree 1 to tree $K$, and for each node we store a value $d_i$ which is the maximum value of paths from tree 1 to this node. At tree $k$ and node $i$, the $d_i$ value can be computed by

$$d_i = v_i + \max_{j:j \in V_{k-1} \text{ and } (j,i) \in E} d_j. \tag{7}$$

Then we take the max $d$ value in the last tree. It produces an upper bound of $v^*$, since the maximum valued path found by dynamic programming is not necessarily a $K$-clique. Again $V_{k-1} = S^{(k-1)}$ in the first level but it will be generalized below.

**Merging $T$ independent sets** To refine the relatively loose single-level bound, we partition the graph into $K/T$ subgraphs, each with $T$ independent sets. Within each subgraph, we find all the $T$-cliques and use a new "pseudo node" to represent each $T$-clique. $T$-cliques in a subgraph can be enumerated efficiently if we choose $T$ to be a relatively small number (e.g., 2 or 3 in the experiments).

Now we exploit the boxicity property to form a new graph among these $T$-cliques (illustrated as the second level nodes in Figure 1). By Lemma 1, we know that the intersection of $T$ boxes will still be

a box, so each $T$-clique is still a box and can be represented as a pseudo node in the level-2 graph. Also because each pseudo node is still a box, we can easily form edges between pseudo nodes to indicate the nonempty overlapping between them and this will be a $(K/T)$-partite boxicity graph since no edge can be formed for the cliques within the same subgraph. Thus we get the level-2 graph. With the level-2 graph, we can again run the single level algorithm to compute a upper bound on $v^*$ to get a lower bound of $r^*$ in (1), but different from the level-1 graph, now we already considered all the within-subgraph edges so the bounds we get will be tighter.

**The overall multi-level framework**   We can run the algorithm level by level until merging all the subgraphs into one, and in the final level the pseudo nodes will correspond to the $K$-cliques in the original graph, and the maximum value will be exactly $v^*$. Therefore, our algorithm can be viewed as an anytime algorithm that refines the upper bound level-by-level until reaching the maximum value. Although getting to the final level still requires exponential time, in practice we can stop at any level (denoted as $L$) and get a reasonable bound. In experiments, we will show that by merging few trees we already get a bound very close to the final solution. Algorithm 2 gives the complete procedure.

---

**Algorithm 2:** Multi-level verification framework

---

**input :** The set of leaf nodes of each tree, $S^{(1)}$, $S^{(2)}$, ..., $S^{(K)}$; maximum number of independent sets in a subgraph (denoted as $T$); maximum number of levels (denoted as $L$), $L \leq \lceil \log_T(K) \rceil$;

1  **for** $k \leftarrow 1, 2, \ldots, K$ **do**
2  $\quad$ $U_k^{(0)} \leftarrow \{(A_i, B^{i^{(k)}}) | i^{(k)} \in S^{(k)}, A_i = \{i^{(k)}\}\}$;
$\quad$ /* $U$ is defined the same as in Algorithm 1. At level 0, each $V_k$ forms a 1-clique by itself.   */
3  **end**
4  **for** $l \leftarrow 1, 2, \ldots, L$ **do**
$\quad$ /* Enumerate all cliques in each subgraph at this level. Total $\lceil K/T^l \rceil$ subgraphs.   */
5  $\quad$ **for** $k \leftarrow 1, 2, \ldots, \lceil K/T^l \rceil$ **do**
6  $\quad\quad$ $U_k^{(l)} \leftarrow \texttt{CliqueEnumerate}(U_{(k-1)T+1}^{(l-1)}, U_{(k-1)T+2}^{(l-1)}, \ldots, U_{kT}^{(l-1)})$;
7  $\quad$ **end**
8  **end**
9  **for** $k \leftarrow 1, 2, \ldots, \lceil K/T^L \rceil$ **do**
$\quad$ /* Define an independent set $V_k'$ for each $U_k^{(L)}$. In each $V_k'$, we create "pseudo nodes" which combines multiple nodes from lower levels, and assign "pseudo prediction values" to them.   */
10  $\quad$ $V_k' \leftarrow \{A \,|\, (A, B) \in U_k^{(L)}\}$;  $\quad$ /* $V_k'$ is a set of sets; each element in $V_k'$ represents a clique.   */
$\quad$ /* Construct the "pseudo prediction value" for each element in $V_k'$ by summing up all prediction values in the corresponding clique.   */
11  $\quad$ For all $A \in V_k'$, $v_A \leftarrow \sum_{i \in A} v_i$
12  **end**
13  $\bar{v} \leftarrow$ an upper bound of $v^*$ using (6) or (7), given $\tilde{V} = \{V_1', \cdots, V_{\lceil K/T^L \rceil}'\}$;
$\quad$ /* If $\lceil K/T^L \rceil = 1$, only 1 independent set left and each pseudo node represents a $K$-clique; (6) or (7) will have a trivial solution where $v^*$ is the maximum $v_A$ in $U_1^{(L)}$.   */

---

**Handling multi-class tree ensembles**   For a multiclass classification problem, say a $C$-class classification problem, $C$ groups of tree ensembles (each with $K$ trees) are built for the classification task; for the $k$-th tree in group $c$, prediction outcome is denoted as $i^{(k,c)} = m^{(k,c)}(x)$ where $m^{(k,c)}(x)$ is the function that maps the input example $x$ to a leaf node of tree $k$ in group $c$. The final prediction is given by $\arg\max_c \sum_k v_{i(k,c)}$. Given an input example $x$ with ground-truth class $c$ and an attack target class $c'$, we extract $2K$ trees for class $c$ and class $c'$, and flip the sign of all prediction values for trees in group $c'$, such that initially $\sum_t v_{i(t,c)} + \sum_t v_{i(t,c')} < 0$ for a correctly classified example. Then, we are back to the binary classification case with $2K$ trees, and we can still apply our multi-level framework to obtain a lower bound $\underline{r}_{(c,c')}$ of $r^*_{(c,c')}$ for this target attack pair $(c, c')$. Robustness of an untargeted attack can be evaluated by taking $\underline{r} = \min_{c' \neq c} \underline{r}_{(c,c')}$.

### 3.4   Verification Problems Beyond Ordinary Robustness

The above discussions focus on the decision problem of $\ell_\infty$ robustness verification (2). In fact, our approach works for a more general verification problem for *any* $d$-dimensional box $B$:

$$\text{Is there any } x' \in B \text{ such that } f(x') \neq y_0? \tag{8}$$

In typical robustness verification settings, $B$ is defined to be $\text{Ball}(x, \epsilon)$ but in fact we can allow any boxes in our algorithm. For a general $B$, Lemma 1 still holds so all of our algorithms and analysis can go through. The only change is to compute the intersection between $B$ and each box of leaf node at the first level in Figure 1 and eliminate nodes that have an empty intersection with $B$. So

robustness verification is just a special case where we remove all the nodes with empty intersection with Ball$(x, \epsilon)$. For example, we can identify a set of unimportant variables, where any individual feature change in this set cannot alter the prediction for a given sample $x$. For each feature $i$, we can choose $B$ as $B_i = [-\infty, \infty]$ (or the the entire input domain, like $[0, 1]$ for image data) and $B_{j \neq i} = \{x_j\}$ otherwise. If the model is robust to such a single-feature perturbation, then this feature is added to the unimportant set. Similarly, we can get a set of anchor features (similar to [30]) such that once a set of features are fixed, any perturbation outside the set cannot change the prediction.

## 4 Experiments

We evaluate our proposed method for robustness verification of tree ensembles on two tasks: binary and multiclass classification on 9 public datasets including both small and large scale datasets. Our code (XGBoost compatible) is available at https://github.com/chenhongge/treeVerification. We run our experiments on Intel Xeon Platinum 8160 CPUs. The datasets other than MNIST and Fashion-MNIST are from LIBSVM [8]. The statistics of the data sets are shown in Appendix A. As we defined in Section 2, $r^*$ is the radius of minimum adversarial perturbation that reflects true model robustness, but is hard to obtain; our method finds $\underline{r}$ that is a lower bound of $r^*$, which guarantees that *no adversarial example* exists within radius $\underline{r}$. A high quality lower bound $\underline{r}$ should be close to $r^*$. We include the following algorithms in our comparisons:

- Cheng's attack [13] provides results on adversarial attacks on these models, which gives an upper bound of the model robustness $r^*$. We denote it as $\overline{r}$ and $\overline{r} \geq r^*$.
- MILP: an MILP (Mixed Integer Linear Programming) based method [20] gives the exact $r^*$. It can be very slow when the number of trees or dimension of the features increases.
- LP relaxation: a Linear Programming (LP) relaxed MILP formulation by directly changing all binary variables to continuous ones. Since the binary constraints are removed, solving the minimization of MILP gives a lower bound of robustness, $\underline{r}_{LP}$, serving as a baseline method.
- Our proposed multi-level verification framework in Section 3.3 (with pseudo code as Algorithm 2 in the appendix). We are targeting to compute robustness interval $\underline{r}_{our}$ for tree ensemble verification.

In Tables 1 and 2 we show empirical comparisons on 9 datasets. We consider $\ell_\infty$ robustness, and normalize our datasets to $[0, 1]$ such that perturbations on different datasets are comparable. We use (6) to obtain single layer bounds. Results using dynamic programming in (7) are provided in Appendix B. We include both standard (naturally trained) GBDT models (Table 1) and robust GBDT models [9] (Table 2). The robust GBDTs were trained by considering model performance under the worst-case perturbation, which leads to a max-min saddle point problem when finding the optimal split at each node [9]. All GBDTs are trained using the XGBoost framework [12]. The number of trees in GBDTs and parameters used in training GBDTs for different datasets are shown in Table 3 in the appendix. Because we solve the decision problem of robustness verification, we use a 10-step binary search to find the largest $\underline{r}$ in all experiments, and the reported time is the total time including all binary search trials. We present the average of $\underline{r}$ or $r^*$ over 500 examples. The MILP based method from [20] is an accurate but very slow method; the results marked with an asterisk ("*") in the table have very long running time and thus we only evaluate 50 examples instead of 500.

| Dataset | Cheng's attack [13] | | MILP [20] | | LP relaxation | | | | Ours (without DP) | | Ours vs. MILP | |
|---|---|---|---|---|---|---|---|---|---|---|---|---|
| | avg. $\overline{r}$ | avg. time | avg. $r^*$ | avg. time | avg. $\underline{r}_{LP}$ | avg. time | $T$ | $L$ | avg. $\underline{r}_{our}$ | avg. time | $\underline{r}_{our}/r^*$ | speedup |
| breast-cancer | .221 | 2.18s | .210 | .012s | .064 | .009s | 2 | 1 | .208 | .001s | .99 | 12X |
| covtype | .058 | 4.76s | .028* | 355*s | .005* | 154*s | 2 | 3 | .022 | 3.39s | .79 | 105X |
| diabetes | .064 | 1.70s | .049 | .061s | .015 | .026s | 3 | 2 | .042 | .018s | .86 | 3.4X |
| Fashion-MNIST | .048 | 12.2s | .014* | 1150*s | .003* | 898*s | 2 | 1 | .012 | 11.8s | .86 | 97X |
| HIGGS | .015 | 3.80s | .0028* | 68*min | .00035* | 50*min | 4 | 1 | .0022 | 1.29s | .79 | 3163X |
| ijcnn1 | .047 | 2.72s | .030 | 4.64s | .008 | 2.67s | 2 | 2 | .026 | .101s | .87 | 4.6X |
| MNIST | .070 | 11.1s | .011* | 367*s | .003* | 332*s | 2 | 2 | .011 | 5.14s | 1.00 | 71X |
| webspam | .027 | 5.83s | .00076 | 47.2s | .0002 | 39.7s | 2 | 1 | .0005 | .404s | .66 | 117X |
| MNIST 2 vs. 6 | .152 | 12.0s | .057 | 23.0s | .016 | 11.6s | 4 | 1 | .046 | .585s | .81 | 39X |

Table 1: Average $\ell_\infty$ distortion over 500 examples and average verification time per example for three verification methods. Here we evaluate the bounds for **standard (natural) GBDT models**. Results marked with a start ("⋆") are the averages of 50 examples due to long running time. $T$ is the number of independent sets and $L$ is the number of levels in searching cliques used in our algorithm. A ratio $\underline{r}_{our}/r^*$ close to 1 indicates better lower bound quality. Dynamic programming in (7) is not applied. Results using dynamic programming are provided in Appendix B.

From Tables 1 and 2 we can see that our method gives a tight lower bound $\underline{r}$ compared to $r^*$ from MILP, while achieving up to $\sim 3000$X speedup on large models. The running time of the baseline

| Dataset | Cheng's attack [13] | | MILP [20] | | LP relaxation | | Ours (without DP) | | | | Ours vs. MILP | |
|---|---|---|---|---|---|---|---|---|---|---|---|---|
| | avg. $\bar{r}$ | avg. time | avg. $r^*$ | avg. time | avg. $\underline{r}_{LP}$ | avg. time | $T$ | $L$ | avg. $\underline{r}_{our}$ | avg. time | $\underline{r}_{our}/r^*$ | speedup |
| breast-cancer | .404 | 1.96s | .400 | .009s | .078 | .008s | 2 | 1 | .399 | .001s | 1.00 | 9X |
| covtype | .079 | .481s | .046* | 305*s | .0053* | 159*s | 2 | 3 | .032 | 4.84s | .70 | 63X |
| diabetes | .137 | 1.52s | .112 | .034s | .035 | .013s | 3 | 2 | .109 | .006s | .97 | 5.7X |
| Fashion-MNIST | .153 | 13.9s | .091* | 41*min | .009* | 34*min | 2 | 1 | .071 | 18.0s | .78 | 137X |
| HIGGS | .023 | 3.58s | .0084* | 59*min | .00031* | 54*min | 4 | 1 | .0063 | 1.41s | .75 | 2511X |
| ijcnn1 | .054 | 2.63s | .036 | 2.52s | .009 | 1.26s | 2 | 2 | .032 | 0.58s | .89 | 4.3X |
| MNIST | .367 | 1.41s | .264* | 615*s | .019* | 515*s | 2 | 2 | .253 | 12.6s | .96 | 49X |
| webspam | .048 | 4.97s | .015 | 83.7s | .0024 | 60.4s | 2 | 1 | .011 | .345s | .73 | 243X |
| MNIST 2 vs. 6 | .397 | 17.2s | .313 | 91.5s | .039 | 40.0s | 4 | 1 | .308 | 3.68s | .98 | 25X |

Table 2: Verification bounds and running time for **robustly trained GBDT models** introduced in [9]. The settings for each method are similar to the settings in Table 1.

LP relaxation, however, is on the same order of magnitude as the MILP method, but the results are much worse, with $\underline{r}_{LP} \ll r^*$. Figure 2 shows how the tightness of our robustness verification lower bounds changes with different size of clique per level ($T$) and different number of levels ($L$). We test on a 20-tree standard GBDT model on the diabetes dataset. We also show the exact bound $r^*$ by the MILP method. Our verification bound converges to the MILP bound as more levels of clique enumerations are used. Also, when we use larger cliques in each level, the bound becomes tighter.

To show the scalability of our method, we vary the number of trees in GBDTs and compare per example running time with the MILP method on ijcnn1 dataset in Figure 3. We see that our multi-level method spends much less time on each example compared to the MILP method and our running time grows slower than MILP when the number of trees increases.

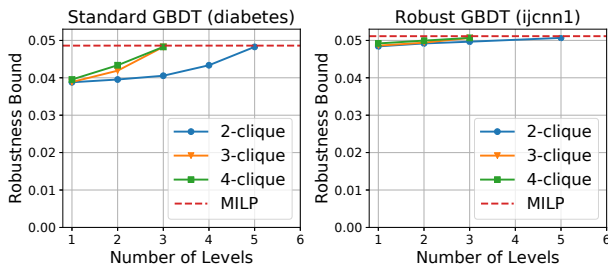
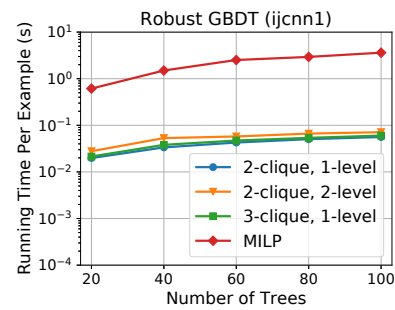

Figure 2: Robustness bounds obtained with different parameters ($T = \{2, 3, 4\}$, $L = \{1, \cdots, 6\}$) on a 20-tree standard GBDT model trained on diabetes dataset (left) and a 20-tree robust GBDT model trained on ijcnn1 dataset (right). $\underline{r}_{our}$ converges to $r^*$ as $L$ increases.

Figure 3: Running time of MILP and our method on robust GBDTs with different number of trees (ijcnn1 dataset).

In Section 3.4, we showed that our algorithm works for more general verification problems such as identifying unimportant features, where any changes on one of those features alone cannot alter the prediction. We use MNIST to demonstrate pixel importance, where we perturb each pixel individually by $\pm\epsilon$ while keeping other pixels unchanged, and obtain the largest $\epsilon$ such that prediction is unchanged. In Figure 4, yellow pixels cannot change prediction for any perturbation and a darker pixel represents a smaller lower bound $\underline{r}$ of perturbation to change the model output using that pixel. The standard naturally trained model has some very dark pixels compared to the robust model. Discussion on the connection between this score and other feature importance scores is in Section C.

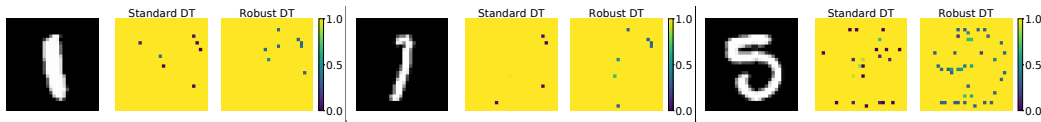

Figure 4: MNIST pixel importance. For each 3-image group, left: digit image; middle: results on standard DT model; right: results on robust DT model. Changing one of any yellow pixels ($\underline{r} = 1.0$) to any valid values between 0 and 1 cannot alter model prediction; pixels in darker colors (smaller $\underline{r}$) tend to affect model prediction more than pixels in lighter colors (larger $\underline{r}$).

**Acknowledgement.** Chen and Boning acknowledge the support of SenseTime. Hsieh acknowledges the support of NSF IIS-1719097 and Intel faculty award.

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
