[Supplementary Material]

## A  Data Statistics and Model Parameters in Tables 1 and 2

Table 3 presents data statistics and parameters for the models in Tables 1 and 2 in the main text. The standard test accuracy is the model accuracy on natural, unmodified test sets.

| Dataset | training set size | test set size | # of features | # of classes | # of trees | robust $\epsilon$ | depth robust | depth natural | standard test acc. robust | natural |
|---|---|---|---|---|---|---|---|---|---|---|
| breast-cancer | 546 | 137 | 10 | 2 | 4 | 0.3 | 8 | 6 | .978 | .964 |
| covtype | 400,000 | 181,000 | 54 | 7 | 80 | 0.2 | 8 | 8 | .847 | .877 |
| diabetes | 614 | 154 | 8 | 2 | 20 | 0.2 | 5 | 5 | .786 | .773 |
| Fashion-MNIST | 60,000 | 10,000 | 784 | 10 | 200 | 0.1 | 8 | 8 | .903 | .903 |
| HIGGS | 10,500,000 | 500,000 | 28 | 2 | 300 | 0.05 | 8 | 8 | .709 | .760 |
| ijcnn1 | 49,990 | 91,701 | 22 | 2 | 60 | 0.1 | 8 | 8 | .959 | .980 |
| MNIST | 60,000 | 10,000 | 784 | 10 | 200 | 0.3 | 8 | 8 | .980 | .980 |
| webspam | 300,000 | 50,000 | 254 | 2 | 100 | 0.05 | 8 | 8 | .983 | .992 |
| MNIST 2 vs. 6 | 11,876 | 1,990 | 784 | 2 | 1000 | 0.3 | 6 | 4 | .997 | .998 |

Table 3: The data statistics and parameters for the models presented in Tables 1 and 2.

## B  Results for Solving Single Layer Bounds with Dynamic Programming

In this section we provide results of our algorithm by using Eq. (7) for solving the last single layer bounds. Since using dynamic programming to find the maximum valued path in a graph can take significantly longer time than using (6), we found that the solving time increases noticeably if using the same $T$ and $L$ values. For some models, we reduce the values of $T$ or $L$ in order to speed up our method with dynamic programming. But even with smaller $T$ or $L$ values, the lower bounds $\underline{r}$ can also be improved with dynamic programming.

| Dataset | MILP [20] avg. $r^*$ | avg. time | $T$ | $L$ | Ours (with DP) avg. $\underline{r}_{our}$ | avg. time | Ours vs. MILP $\underline{r}_{our}/r^*$ | speedup |
|---|---|---|---|---|---|---|---|---|
| breast-cancer | .210 | .012s | 2 | 1 | .209 | .001s | 1.00 | 12X |
| covtype | .028* | 355*s | 2 | 3 | .024 | 5.70s | .86 | 62X |
| diabetes | .049 | .061s | 2 | 2 | .044 | .013s | .90 | 4.7X |
| Fashion-MNIST | .014* | 1150*s | 2 | 1 | .012 | 22.8s | .86 | 50X |
| HIGGS | .0028* | 68*min | 4 | 1 | .0023 | 22.1s | .82 | 185X |
| ijcnn1 | .030 | 4.64s | 2 | 1 | .027 | .053s | .90 | 88X |
| MNIST | .011* | 367*s | 2 | 1 | .011 | 5.10s | 1.00 | 72X |
| webspam | .00076 | 47.2s | 2 | 1 | .00051 | 3.29s | .67 | 14X |
| MNIST 2 vs. 6 | .057 | 23.0s | 4 | 1 | .050 | 2.41s | .88 | 9.5X |

Table 4: Average $\ell_\infty$ distortion over 500 examples and average verification time per example for three verification methods. Here we evaluate the bounds for **standard (natural) GBDT models**. Results marked with a star ("$\star$") are the averages of 50 examples due to long running time. $T$ is the number of independent sets and $L$ is the number of levels in searching cliques used in our algorithm. A ratio $\underline{r}_{our}/r^*$ close to 1 indicates better lower bound quality.

| Dataset | MILP [20] avg. $r^*$ | avg. time | $T$ | $L$ | Ours (with DP) avg. $\underline{r}_{our}$ | avg. time | Ours vs. MILP $\underline{r}_{our}/r^*$ | speedup |
|---|---|---|---|---|---|---|---|---|
| breast-cancer | .400 | .009s | 2 | 1 | .399 | .001s | 1.00 | 9.0X |
| covtype | .046* | 305*s | 2 | 2 | .035 | 3.69s | .76 | 83X |
| diabetes | .112 | .034s | 2 | 2 | .111 | .005s | .98 | 7.1X |
| Fashion-MNIST | .091* | 41*min | 2 | 1 | .071 | 19.9s | .78 | 124X |
| HIGGS | .0084* | 59*min | 4 | 1 | .0069 | 4.25s | .82 | 783X |
| ijcnn1 | .036 | 2.52s | 2 | 2 | .035 | .655s | .97 | 3.8X |
| MNIST | .264* | 615*s | 2 | 1 | .264 | 7.74s | 1.00 | 63X |
| webspam | .015 | 83.7s | 2 | 1 | .011 | 1.26s | .73 | 66X |
| MNIST 2 vs. 6 | .313 | 91.5s | 2 | 1 | .309 | 5.91s | .99 | 15.5X |

Table 5: Verification bounds and running time for **robustly trained GBDT models** introduced in [9]. The settings for each method are similar to the settings in Table 4.

## C  Connection between the Score in Figure 4 and Other Feature Importance Scores

We note that our perturbation-sensitivity notion of feature importance is complementary to the conventional tree/forest feature importance, with several critical differences. In Figure 5 below we show the feature importance map of the same standard and robust models used in Figure 4 in the main text. A feature's importance is measured by the average gain across all the splits it is used in. Pixels with darker color have larger importance and yellow pixels have zero importance. Our single-feature robustness bounds shown in Figure 4 are different from importance scores (Figure 5) in the following ways:

- The conventional feature importance score only depends on the model itself, and is test data independent. Conversely, our single-feature robustness bound depends on both the model and the test data point; for different data points, the model may be sensitive to different features.

- The conventional feature importance is a heuristic score. Our robustness bound can give a formal guarantee that the model output would not change if this single feature is perturbed within a given range.
- The conventional feature importance score assigns non-zero importance to more pixels than our method does in general.

Figure 5: Feature importance of the same models as in Figure 4 in the main text. Left: standard DT model; Right: robust DT model. Yellow pixels have zero feature importance while darker pixels have larger importance. A feature's importance is measured by the average gain across all the splits it is used in.

## D  Proof of Lemma 1

**Lemma 1.** For boxes $B^1, \ldots, B^K$, if $B^i \cap B^j \neq \emptyset$ for all $i, j \in [K]$, let $\bar{B} = B^1 \cap B^2 \cap \cdots \cap B^K$ be their intersection. Then $\bar{B}$ will also be a box and $\bar{B} \neq \emptyset$.

*Proof.* If we have $K$ one dimensional intervals $I_1 = (l_1, r_1], I_2 = (l_2, r_2], \ldots, I_T = (l_K, r_K]$, we want to prove if every pair of them have nonempty overlap $I_1 \cap \cdots \cap I_K \neq \emptyset$. This can be proved by the following. Without loss of generality we assume $l_1 \leq l_2 \leq \cdots \leq l_K$. For each $k < K$, $I_k \cap I_K \neq \emptyset$ implies $l_K < r_k$. Therefore, $(l_T, \min(r_1, r_2, \ldots, r_K)]$ will be a nonempty set that is contained in $I_1, I_2, \ldots, I_K$. Therefore $I_1 \cap I_2 \cap \cdots \cap I_K \neq \emptyset$ and it is another interval.

This can be generalized to $d$-dimensional boxes. Assume we have boxes $B_1, \ldots, B_K$ such that $B_i \cap B_j \neq \emptyset$ for any $i$ and $j$. Then for each dimension we can apply the above proof, which implies that $B_1 \cap B_2 \cap \cdots \cap B_K \neq \emptyset$ and the intersection will be another box. □

## E  An $O(n)$ time algorithm for verifying a decision tree

The robustness of a single tree can be easily verified by the following $O(n)$ algorithm, which traverse the whole tree and computes the bounding boxes for each node in a depth-first search fashion.

---
**Algorithm 3:** Linear time $\ell_\infty$ untargeted attack for a decision tree.

---

```
1  Initial p* = 0, ℓ_t = −∞, r_t = ∞,  ∀t = 1, . . . d;
2  ComputeRecursive(0, 0);

3  Function ComputeRecursive(i, p)
4      if i is leaf node then
5          if v_i ≠ y_0 then
6              │  p* ← min(p*, p);
7      else
           /* Checking conditions for the left child                                    */
8          s ← r_{t_i} ;
9          r_{t_i} ← min(r_{t_i}, l_{t_i}) ;
10         if l_{t_i} ≤ r_{t_i} then
11             if r_{t_i} < x_{t_i} then
12                 │  ComputeRecursive(i.left_child, max(p, |x_{t_i} − r_{t_i}|))
13             else
14                 │  ComputeRecursive(i.left_child, p) ;
15         r_{ti} ← s;
           /* Checking conditions for the right child                                   */
16         s ← l_{t_i} ;
17         l_{t_i} ← max(l_{t_i}, l_{t_i}) ;
18         if l_{t_i} ≤ r_{t_i} then
19             if l_{t_i} > x_{t_i} then
20                 │  ComputeRecursive(i.right_child, max(p, |x_{t_i} − l_{t_i}|))
21             else
22                 │  ComputeRecursive(i.right_child, p) ;
23  end
```

---