[Reviews · NeurIPS 2019]

Reviewer 1



Originality: The robustness verification methods presented in the paper is new and interesting. The authors provided a fair list of related work and compared the existing methods with their method in the experiment section. Quality: The paper provides a complete presentation of three verification methods, 1) verifying the robustness of a single decision tree, 2) verifying the robustness of a tree ensemble using existing algorithms for finding k-cliques, and 3) a fast and approximate method for estimating a lower bound on the robustness. The theoretical claims and their proofs make sense to me. Overall the empirical evaluation is well designed and convincing. I only have two questions: 1. What are the ensemble sizes for the models trained on the datasets presented in Table 1 and 2? 2. How much does the parameters T and L effect the running time? Clarity: This paper is well written and well organized. The authors did a good work on describing the objective of their method, and provided a fair amount of introduction on the background and related work. It would be desirable if the authors consider releasing the source code of their method, which is not provided with this submission (though pseudo code is provided in the appendix). Significance: The robustness of the tree models are attracting increasing attention due to their popularity in the real-world applications. This paper presents an effective method for quantifying the robustness of tree ensembles. Empirical evaluation shows that the proposed method can provide evaluation in seconds per example, which is often order of magnitude faster than the existing methods. Overall I think this paper is valuable for the practitioner for evaluating the robustness of the tree ensembles. --- Update: The authors clarified my questions. I also found the answer to measuring the feature importance (a question from Reviewer #2) could be a very good addition to the text. Overall, I maintain my original assessment that this is a good submission.

Reviewer 2



Novelty: The theoretical results and the algorithm are new, and it's clear how they are different from the related work cited in the paper. It's also nice that they reduced this problem to a well-studied one. Quality: Overall, the paper is well put together, in that it presents a problem that was recognized as important, followed by a formalization and solution in a simple case, then the main theoretical result which appears to be sound, then a pertinent algorithm which provides fast solutions, and finally an evaluation of this algorithm in terms of speed and minimal adversarial perturbation. That the procedure can also be used to find features that are unimportant and 'immune' to perturbation is also a point it its favour, tough wouldn't we just be able to identify these by calculating variable importance in a forest? There is one point that I'm not convinced about: the authors rely on the conclusion by previous work that adversarial attacks can have a significant influence on model performance (references 8,12,18). [12] introduces a new type of attack, which as far as I can tell does not affect the results in this paper. However, [8] and [18] introduce methods for strengthening ensembles against adversarial perturbations. How do the bounds hold if these methods are applied instead of standard trees, which is the case for any application where adversarial attacks are expected? The algorithm is applied on [8] (robust GBDT) which is the more recent work. It does seem to be the case that the bounds are much closer to the MILP ones in this case (Fig 2 right). Clarity: The paper is clear and well organized. The main paper introduces enough information for an implementation of the algorithm, but the appendix would be needed to actually replicate the results. Update following the author response: The authors answered my comments. I liked the feature importance example - perhaps it could be included in the appendix?

Reviewer 3



This paper studied the robustness verification problem for tree-based model. Although formal robustness verification is NP-complete, they give a more precise complexity characterization. For general problems, by exploiting the boxicity of the graph, they develop an efficient multi-level verification algorithm. The idea is novel and interesting. The proposed algorithm can give tight lower bounds on robustness of decision tree ensembles, while allowing iterative improvement and any-time termination. Experiments on 10 datasets show the superior performance of the proposed method. This paper is well written and easy to follow.

[Author Response · NeurIPS 2019]

We thank all three reviewers for their constructive comments. Our responses are as follows:

**Reviewer 1.**

1. The parameters (including ensemble sizes) for the models are presented in Table 3 in Appendix B in the supplemental materials.

2. Figure 3 in the main text shows the running time of MILP and our method on GBDT models with different T, L and numbers of trees on the ijcnn1 dataset. There, we see that T and L have a modest impact on running time, with a larger (though still slower than MILP) impact from the number of trees. We see that T (clique size per level) has a smaller impact on running time than L (the number of levels).

3. Thank you for the question and reminder to upload our code. To comply with NeurIPS rules, we have sent an anonymous GitHub link with our code to AC and will let AC decide whether to release it to reviewers.

**Reviewer 2.**

1. We note that our perturbation-sensitive notion of feature importance is complementary to the conventional tree/forest feature importance, with several critical differences. In Figure 1 below we show the feature importance map of the standard and robust models used in Figure 4 in the main text. A feature's importance is measured by the average gain across all the splits it is used in. Pixels with darker color have larger importance and yellow pixels have zero importance. Our single-feature robustness bounds for different features shown in Figure 4 are different from importance scores in the following ways:

- Feature importance scores only depend on the model itself. But our single-feature robustness bound depends on both the model and the testing data point, and for different data points, the model may be sensitive to different features.
- Feature importance is a relatively heuristic score. But our robustness bound can give a formal guarantee that the model output would not change if this single feature is perturbed within the given range.
- A feature importance score assigns non-zero importance to more pixels than our method in general.

Figure 1: Feature importance of the model in Figure 4 in the main text. Left: standard GBDT model; Right: robust GBDT model. Yellow pixels have zero feature importance while darker pixels have larger importance.

2. The reviewer asks about the relationship of different attacks and methods seeking to improve robustness ([8], [12], [18]), with respect to impacts on our verification bounds. In response, we first confirm that, as suggested in [8], adversarial examples indeed exist in tree-based models. In Figure 1 of [8] the authors used a black-box attack developed in [12] to generate adversarial examples on MNIST and Fashion-MNIST. The noise added to generate these adversarial examples are so small that is indistinguishable for human eyes but can completely fool the model. Authors in [18] proposed an even stronger MILP based white-box attack designed specifically for GBDT models to find the best adversarial example with the exact minimum distortion. The robust training strategies introduced in [8] and [18] can indeed increase model robustness, and thus our verification bounds also increase in value correspondingly. In this case, our bounds are still a lower bound with respect to the exact MILP bound. Note that our verification bound is used to evaluate model robustness, instead of attacking models. Table 2 shows our bounds applied to robust models trained using [8] and the bounds indeed increase compared to standard model bounds in Table 1. The tightness of the bound depends on the model, and one cannot guarantee that robust model bounds are tighter than of a standard model. For example, in Tables 1 and 2, on covtype dataset, the standard model's bound is tighter than robust model's bound when we use the same T and L values. However, intuitively, the robust training method in [8] has a regularization effect, so the model tends to be sparser with fewer boxes and simpler overlapping structures. We believe that this also contributes to robustness of the models and tighter bounds.

3. MILP is also an adversarial attack method, and the bound obtained by MILP is the **exact** minimum distortion with formal mathematical guarantees. Therefore MILP is used as a ground truth benchmark to test our verification bounds, and it is not possible to obtain a tighter bound than it. However, MILP can be very slow in large models. Our bound is much faster than MILP, and very tight compared to it as shown in the experiments.

**Reviewer 3.**

1. $\ell_\infty$ balls are larger than all $\ell_p$ balls with the same radius. Since our current bound can guarantee that no adversarial example exists in an $\ell_\infty$ ball with radius $R$, we can also guarantee that no adversarial example exists in any $\ell_p$ balls with the same radius. We will leave the development of bounds designed specifically for other $\ell_p$ norms for future studies.

2. If you mean "why LP relaxation bound is much worse than MILP and our bound?", the reason is that LP relaxation relaxes all 0-1 constraints in MILP to [0,1] and the search space is much larger than MILP. Therefore a much smaller distortion is found by LP relaxation. However, since this relaxation is so loose, the data point found by LP relaxation is very likely not an adversarial example.

[Meta-Review · NeurIPS 2019]

This paper studies the problem of robustness verification for an ensemble of K trees. The authors reduce this problem to maximum clique on a k-partite graph which is interesting. They also show that the proposed algorithm is faster compared to a generic integer programming approach (which was proposed previously for robustness).